# On the Optimality of Perturbations in Stochastic and Adversarial Multi-armed Bandit Problems

**Baekjin Kim**
Department of Statistics
University of Michigan
Ann Arbor, MI 48109
`baekjin@umich.edu`

**Ambuj Tewari**
Department of Statistics
University of Michigan
Ann Arbor, MI 48109
`tewaria@umich.edu`

## Abstract

We investigate the optimality of perturbation based algorithms in the stochastic and adversarial multi-armed bandit problems. For the stochastic case, we provide a unified regret analysis for both sub-Weibull and bounded perturbations when rewards are sub-Gaussian. Our bounds are instance optimal for sub-Weibull perturbations with parameter 2 that also have a matching lower tail bound, and all bounded support perturbations where there is sufficient probability mass at the extremes of the support. For the adversarial setting, we prove rigorous barriers against two natural solution approaches using tools from discrete choice theory and extreme value theory. Our results suggest that the optimal perturbation, if it exists, will be of Fréchet-type.

## 1 Introduction

Beginning with the seminal work of Hannan [12], researchers have been interested in algorithms that use *random perturbations* to generate a distribution over available actions. Kalai and Vempala [17] showed that the perturbation idea leads to efficient algorithms for many online learning problems with large action sets. Due to the *Gumbel lemma* [14], the well known exponential weights algorithm [11] also has an interpretation as a perturbation based algorithm that uses Gumbel distributed perturbations.

There have been several attempts to analyze the regret of perturbation based algorithms with specific distributions such as Uniform, Double-exponential, drop-out and random walk (see, e.g., [17, 18, 9, 28]). These works provided rigorous guarantees but the techniques they used did not generalize to general perturbations. Recent work [1] provided a general framework to understand general perturbations and clarified the relation between regularization and perturbation by understanding them as different ways to smooth an underlying non-smooth potential function.

Abernethy et al. [2] extended the analysis of general perturbations to the partial information setting of the adversarial multi-armed bandit problem. They isolated *bounded hazard rate* as an important property of a perturbation and gave several examples of perturbations that lead to the near optimal regret bound of $O(\sqrt{KT \log K})$. Since Tsallis entropy regularization can achieve the minimax regret of $O(\sqrt{KT})$ [4, 5], the question of whether perturbations can match the power of regularizers remained open for the adversarial multi-armed bandit problem.

In this paper, we build upon previous works [1, 2] in two distinct but related directions. First, we provide the first general result for perturbation algorithms in the *stochastic* multi-armed bandit problem. We present a unified regret analysis for both sub-Weibull and bounded perturbations when rewards are sub-Gaussian. Our regrets are instance optimal for sub-Weibull perturbations with parameter 2 (with a matching lower tail bound), and all bounded support perturbations where there is sufficient probability mass at the extremes of the support. Since the Uniform and Rademacher

distribution are instances of these bounded support perturbations, one of our results is a regret bound for a randomized version of UCB where the algorithm picks a random number in the confidence interval or randomly chooses between lower and upper confidence bounds instead of always picking the upper bound. Our analysis relies on the simple but powerful observation that Thompson sampling with Gaussian priors and rewards can also be interpreted as a perturbation algorithm with Gaussian perturbations. We are able to generalize both the upper bound and lower bound of Agrawal and Goyal [3] in two respects; (1) from the special Gaussian perturbation to general sub-Weibull or bounded perturbations, and (2) from the special Gaussian rewards to general sub-Gaussian rewards.

Second, we return to the open problem mentioned above: is there a perturbation that gives us minimax optimality? We do not resolve it but provide rigorous proofs that *there are barriers to two natural approaches to solving the open problem.* (A) One cannot simply find a perturbation that is exactly equivalent to Tsallis entropy. This is surprising since Shannon entropy does have an exact equivalent perturbation, viz. Gumbel. (B) One cannot simply do a better analysis of perturbations used before [2] and plug the results into their general regret bound to eliminate the extra $O(\sqrt{\log K})$ factor. In proving the first barrier, we use a fundamental result in discrete choice theory. For the second barrier, we rely on tools from extreme value theory.

## 2 Problem Setup

In every round $t$ starting at 1, a learner chooses an action $A_t \in [K] \triangleq \{1, 2, \cdots, K\}$ out of $K$ arms and the environment picks a response in the form of a real-valued reward vector $\mathbf{g}_t \in [0, 1]^K$. While the entire reward vector $\mathbf{g}_t$ is revealed to the learner in the full information setting, the learner only receives a reward associated with his choice in the bandit setting, while any information on other arms is not provided. Thus, we denote the reward corresponding to his choice $A_t$ as $X_t = g_{t, A_t}$.

In stochastic multi-armed bandit, the rewards $g_{t,i}$ are sampled i.i.d from a fixed, but unknown distribution with mean $\mu_i$. Adversarial multi-armed bandit is more general in that all assumptions on how rewards are assigned to arms are dropped. It only assumes that rewards are assigned by an adversary before the interaction begins. Such an adversary is called an *oblivious adversary*. In both environments, the learner makes a sequence of decisions $A_t$ based on each history $\mathcal{H}_{t-1} = (A_1, X_1, \cdots, A_{t-1}, X_{t-1})$ to maximize the cumulative reward, $\sum_{t=1}^{T} X_t$.

As a measure of evaluating a learner, *Regret* is the difference between rewards the learner would have received had he played the best in hindsight, and the rewards he actually received. Therefore, minimizing the regret is equivalent to maximizing the expected cumulative reward. We consider the expected regret, $\mathrm{R}(T) = \mathrm{E}[\max_{i \in [K]} \sum_{t=1}^{T} g_{t,i} - \sum_{t=1}^{T} g_{t,A_t}]$ in adversarial setting, and the pseudo regret, $\mathrm{R}'(T) = T \cdot \max_{i \in [K]} \mu_i - \mathrm{E}[\sum_{t=1}^{T} X_t]$ in stochastic setting. Note that two regrets are the same where an oblivious adversary is considered. An online algorithm is called a *no-regret algorithm* if for every adversary, the expected regret with respect to every action $A_t$ is sub-linear in $T$.

We use FTPL (Follow The Perturbed Leader) to denote families of algorithms for both stochastic and adversarial settings. The common core of FTPL algorithms consists in adding random perturbations to the estimates of rewards of each arm prior to computing the current "the best arm" (or "leader"). However, the estimates used are different in the two settings: stochastic setting uses sample means and adversarial setting uses inverse probability weighted estimates.

## 3 Stochastic Bandits

In this section, we propose FTPL algorithms for stochastic multi-armed bandits and characterize a family of perturbations that make the algorithm instance-optimal in terms of regret bounds. This work is mainly motivated by Thompson Sampling [25], one of the standard algorithms in stochastic settings. We also provide a lower bound for the regret of this FTPL algorithm.

For our analysis, we assume, without loss of generality, that arm 1 is optimal, $\mu_1 = \max_{i \in [K]} \mu_i$, and the sub-optimality gap is denoted as $\Delta_i = \mu_1 - \mu_i$. Let $\hat{\mu}_i(t)$ be the average reward received from arm $i$ after round $t$ written formally as $\hat{\mu}_i(t) = \sum_{s=1}^{t} \mathrm{I}\{A_s = i\} X_s / T_i(t)$ where $T_i(t) = \sum_{s=1}^{t} \mathrm{I}\{A_s = i\}$ is the number of times arm $i$ has been pulled after round $t$. The regret for stochastic

bandits can be decomposed into $R(T) = \sum_{i=1}^{K} \Delta_i E[T_i(T)]$. The reward distributions are generally assumed to be sub-Gaussian with parameter 1 [20].

**Definition 1** (sub-Gaussian). *A random variable $Z$ with mean $\mu = E[Z]$ is sub-Gaussian with parameter $\sigma > 0$ if it satisfies $P(|Z - \mu| \geq t) \leq \exp(-t^2/(2\sigma^2))$ for all $t \geq 0$.*

**Lemma 1** (Hoeffding bound of sub-Gaussian [15]). *Suppose $Z_i$, $i \in [n]$ are i.i.d. random variables with $E(Z_i) = \mu$ and sub-Gaussian with parameter $\sigma$. Then $P(\bar{Z}_n - \mu \geq t) \vee P(\bar{Z}_n - \mu \leq -t) \leq \exp(-nt^2/(2\sigma^2))$ for all $t \geq 0$, where $\bar{Z}_n = \sum_{i=1}^{n} Z_i/n$.*

## 3.1 Upper Confidence Bound and Thompson Sampling

The standard algorithms in stochastic bandit are Upper Confidence Bound (UCB1) [6] and Thompson Sampling [25]. The former algorithm is constructed to compare the largest plausible estimate of mean for each arm based on the optimism in the face of uncertainty so that it would be deterministic in contradistinction to the latter one. At time $t + 1$, UCB1 chooses an action $A_{t+1}$ by maximizing upper confidence bounds, $UCB_i(t) = \hat{\mu}_i(t) + \sqrt{2 \log T/T_i(t)}$. Regarding the instance-dependent regret of UCB1, there exists some universal constant $C > 0$ such that $R(T) \leq C \sum_{i:\Delta_i>0}(\Delta_i + \log T/\Delta_i)$.

Thompson Sampling is a Bayesian solution based on randomized probability matching approach [24]. Given the prior distribution $Q_0$, at time $t + 1$, it computes posterior distribution $Q_t$ based on observed data, samples $\nu_t$ from posterior $Q_t$, and then chooses the arm $A_{t+1} = \arg\max_{i \in [k]} \mu_i(\nu_t)$. In Gaussian Thompson Sampling where the Gaussian rewards $\mathcal{N}(\mu_i, 1)$ and the Gaussian prior distribution for each $\mu_i$ with mean $\mu_0$ and infinite variance are considered, the policy from Thompson Sampling is to choose an index that maximizes $\theta_i(t)$ randomly sampled from Gaussian posterior distribution, $\mathcal{N}(\hat{\mu}_i(t), 1/T_i(t))$ as stated in Alg.1-(i). Also, its regret bound is restated in Theorem 2.

---

**Initialize** $T_i(0) = 0, \hat{\mu}_i(0) = 0$ for all $i \in [K]$
**for** $t = 1$ **to** $T$ **do**
    For each arm $i \in [K]$,
        **(i) Gaussian Thompson Sampling :** $\theta_i(t-1) \sim \mathcal{N}\big(\hat{\mu}_i(t-1), \frac{1}{1 \vee T_i(t-1)}\big)$
        **(ii) FTPL via Unbounded Perturbation :** $\theta_i(t-1) = \hat{\mu}_i(t-1) + \frac{1}{\sqrt{1 \vee T_i(t-1)}} \cdot Z_{it}$
        where $Z_{it}$s are randomly sampled from unbounded $Z$.
        **(iii) FTPL via Bounded Perturbation :** $\theta_i(t-1) = \hat{\mu}_i(t-1) + \sqrt{\frac{(2+\epsilon) \log T}{1 \vee T_i(t-1)}} \cdot Z_{it}$
        where $Z_{it}$s are randomly sampled from $Z \in [-1, 1]$.
    Learner chooses $A_t = \arg\max_{i \in [K]} \theta_i(t-1)$ and receives the reward of $X_t \in [0, 1]$.
    Update : $\hat{\mu}_{A_t}(t) = \frac{\hat{\mu}_{A_t}(t-1) \cdot T_{A_t}(t-1) + X_t}{T_{A_t}(t-1)+1}, T_{A_t}(t) = T_{A_t}(t-1) + 1$.
**end**

**Algorithm 1:** Randomized probability matching algorithms via Perturbation

---

**Theorem 2** (Theorem 3 [3]). *Assume that reward distribution of each arm $i$ is Gaussian with mean $\mu_i$ and unit variance. Thompson sampling policy via Gaussian prior defined in Alg.1-(i) has the following instance-dependent and independent regret bounds, for $C' > 0$,*

$$R(T) \leq C' \sum_{\Delta_i>0} \Big(\log(T\Delta_i^2)/\Delta_i + \Delta_i\Big), \quad R(T) \leq \mathcal{O}(\sqrt{KT \log K}).$$

**Viewpoint of Follow-The-Perturbed-Leader**    The more generic view of Thompson Sampling is via the idea of perturbation. We bring an interpretation of viewing this Gaussian Thompson Sampling as Follow-The-Perturbed-Leader (FTPL) algorithm via Gaussian perturbation [20]. If Gaussian random variables $\theta_i(t)$ are decomposed into the average mean reward of each arm $\hat{\mu}_i(t)$ and scaled Gaussian perturbation $\eta_{it} \cdot Z_{it}$ where $\eta_{it} = 1/\sqrt{T_i(t)}, Z_{it} \sim N(0, 1)$. In a round $t + 1$, the FTPL algorithm chooses the action according to $A_{t+1} = \arg\max_{i \in [K]} \hat{\mu}_i(t) + \eta_{it} \cdot Z_{it}$.

## 3.2 Follow-The-Perturbed-Leader

We show that the FTPL algorithm with Gaussian perturbation under Gaussian reward setting can be extended to sub-Gaussian rewards as well as families of sub-Weibull and bounded perturbations. The

sub-Weibull family is an interesting family in that it includes well known families like sub-Gaussian and sub-Exponential as special cases. We propose perturbation based algorithms via sub-Weibull and bounded perturbation in Alg.1-(ii), (iii), and their regrets are analyzed in Theorem 3 and 5.

**Definition 2** (sub-Weibull [29]). *A random variable $Z$ with mean $\mu = \mathrm{E}[Z]$ is sub-Weibull (p) with $\sigma > 0$ if it satisfies $\mathrm{P}(|Z - \mu| \geq t) \leq C_a \exp(-t^p/(2\sigma^p))$ for all $t \geq 0$.*

**Theorem 3** (FTPL via sub-Weibull Perturbation, Proof in Appendix A.1). *Assume that reward distribution of each arm $i$ is 1-sub-Gaussian with mean $\mu_i$, and the sub-Weibull (p) perturbation $Z$ with parameter $\sigma$ and $\mathrm{E}[Z] = 0$ satisfies the following anti-concentration inequality,*

$$\mathrm{P}(|Z| \geq t) \geq \exp(-t^q/2\sigma^q)/C_b, \quad for\ t \geq 0 \tag{1}$$

*Then the Follow-The-Perturbed-Leader algorithm via $Z$ in Alg.1-(ii) has the following instance-dependent and independent regret bounds, for $p \leq q \leq 2$ (if $q = 2$, $\sigma \geq 1$) and $C'' = C(\sigma, p, q) > 0$,*

$$\mathrm{R}(T) \leq C'' \sum_{\Delta_i > 0} \left( \left[ \log(T\Delta_i^2) \right]^{2/p} / \Delta_i + \Delta_i \right), \quad \mathrm{R}(T) \leq \mathcal{O}(\sqrt{KT}(\log K)^{1/p}). \tag{2}$$

Note that the parameters $p$ and $q$ can be chosen from any values $p \leq q \leq 2$, and the algorithm can achieve smaller regret bound as $p$ becomes larger. For nice distributions such as Gaussian and Double-exponential, the parameters $p$ and $q$ can be matched by 2 and 1, respectively.

**Corollary 4** (FTPL via Gaussian Perturbation). *Assume that reward distribution of each arm $i$ is 1-sub-Gaussian with mean $\mu_i$. The Follow-The-Perturbed-Leader algorithm via Gaussian perturbation $Z$ with parameter $\sigma$ and $\mathrm{E}[Z] = 0$ in Alg.1-(ii) has the following instance-dependent and independent regret bounds, for $C'' = C(\sigma, 2, 2) > 0$ and $\sigma \geq 1$,*

$$\mathrm{R}(T) \leq C'' \sum_{\Delta_i > 0} \left( \log(T\Delta_i^2)/\Delta_i + \Delta_i \right), \quad \mathrm{R}(T) \leq \mathcal{O}(\sqrt{KT \log K}). \tag{3}$$

**Failure of Bounded Perturbation** Any perturbation with bounded support cannot yield an optimal FTPL algorithm. For example, in a two-armed bandit setting with $\mu_1 = 1$ and $\mu_2 = 0$, rewards of each arm $i$ are generated from Gaussian distribution with mean $\mu_i$ and unit variance and perturbation is uniform with support $[-1, 1]$. In the case where we have $T_1(10) = 1, T_2(10) = 9$ during first 10 times, and average mean rewards are $\hat{\mu}_1 = -1$ and $\hat{\mu}_2 = 1/3$, then perturbed rewards are sampled from $\theta_1 \sim \mathcal{U}[-2, 0]$ and $\theta_2 \sim \mathcal{U}[0, 2/3]$. This algorithm will not choose the first arm and accordingly achieve a linear regret. To overcome this limitation of bounded support, we suggest another FTPL algorithm via bounded perturbation by adding an extra logarithmic term in $T$ as stated in Alg.1-(iii).

**Theorem 5** (FTPL algorithm via Bounded support Perturbation, Proof in Appendix A.3). *Assume that reward distribution of each arm $i$ is 1-sub-Gaussian with mean $\mu_i$, the perturbation distribution $Z$ with $\mathrm{E}[Z] = 0$ lies in $[-1, 1]$ and for any $\epsilon > 0$, there exists $0 < M_{Z,\epsilon} < \infty$ s.t. $\mathrm{P}\left(Z \leq \sqrt{2/(2+\epsilon)}\right)/\mathrm{P}\left(Z \geq \sqrt{2/(2+\epsilon)}\right) = M_{Z,\epsilon}$. Then the Follow-The-Perturbed-Leader algorithm via $Z$ in Alg.1-(iii) has the following instance-dependent and independent regret bounds, for $C''' > 0$ independent of $T, K$ and $\Delta_i$,*

$$\mathrm{R}(T) \leq C''' \sum_{\Delta_i > 0} \left( \log(T)/\Delta_i + \Delta_i \right), \quad \mathrm{R}(T) \leq \mathcal{O}(\sqrt{KT \log T}). \tag{4}$$

**Randomized Confidence Bound algorithm** Theorem 5 implies that the optimism embedded in UCB can be replaced by simple randomization. Instead of comparing upper confidence bounds, our modification is to compare a value randomly chosen from confidence interval or between lower and upper confidence bounds by introducing uniform $\mathcal{U}[-1, 1]$ or Rademacher perturbation $\mathcal{R}\{-1, 1\}$ in UCB1 algorithm with slightly wider confidence interval, $A_{t+1} = \arg\max_{i \in [K]} \hat{\mu}_i(t) + \sqrt{(2+\epsilon) \log T/T_i(t)} \cdot Z_{it}$. These FTPL algorithms via Uniform and Rademacher perturbations can be regarded as a randomized version of UCB algorithm, which we call the RCB (Randomized Confidence Bound) algorithm, and they also achieve the same regret bound as that of UCB1. The RCB algorithm is meaningful in that it can be arrived at from two different perspectives, either as a randomized variant of UCB or by replacing the Gaussian distribution with Uniform in Gaussian Thompson Sampling.

The regret lower bound of the FTPL algorithm in Alg.1-(ii) is built on the work of Agrawal and Goyal [3]. Theorem 6 states that the regret lower bound depends on the lower bound of the tail probability

of perturbation. As special cases, FTPL algorithms via Gaussian ($q = 2$) and Double-exponential ($q = 1$) make the lower and upper regret bounds matched, $\Theta(\sqrt{KT}(\log K)^{1/q})$.

**Theorem 6** (Regret lower bound, Proof in Appendix A.4)**.** *If the perturbation $Z$ with $\mathrm{E}[Z] = 0$ has the lower bound of tail probability as $\mathrm{P}(|Z| \geq t) \geq \exp[-t^q/(2\sigma^q)]/C_b$ for $t \geq 0, \sigma > 0$, the Follow-The-Perturbed-Leader algorithm via $Z$ has the lower bound of expected regret, $\Omega(\sqrt{KT}(\log K)^{1/q})$.*

## 4   Adversarial Bandits

In this section we study two major families of online learning, Follow The Regularized Leader (FTRL) and Follow The Perturbed Leader (FTPL), as ways of smoothings and introduce the Gradient Based Prediction Algorithm (GBPA) family for solving the adversarial multi-armed bandit problem. Then, we mention an important open problem regarding existence of an optimal FTPL algorithm. The main contributions of this section are theoretical results showing that two natural approaches to solving the open problem are not going to work. We also make some conjectures on what alternative ideas might work.

### 4.1   FTRL and FTPL as Two Types of Smoothings and An Open Problem

Following previous work [2], we consider a general algorithmic framework, Alg.2. There are two main ingredients of GBPA. The first ingredient is the smoothed potential $\tilde{\Phi}$ whose gradient is used to map the current estimate of the cumulative reward vector to a probability distribution $\mathbf{p}_t$ over arms. The second ingredient is the construction of an unbiased estimate $\hat{\mathbf{g}}_t$ of the rewards vector using the reward of the pulled arm only by inverse probability weighting. This step reduces the bandit setting to full-information setting so that any algorithm for the full-information setting can be immediately applied to the bandit setting.

---

**GBPA($\tilde{\Phi}$)**: $\tilde{\Phi}$ differentiable convex function s.t. $\nabla\tilde{\Phi} \in \Delta_{K-1}$ and $\nabla_i\tilde{\Phi} > 0, \forall i$. Initialize $\hat{\mathbf{G}}_0 = \mathbf{0}$.
**for** $t = 1$ *to* $T$ **do**

  A reward vector $\mathbf{g}_t \in [0, 1]^K$ is chosen by environment.
  Learner chooses $A_t$ randomly sampled from the distribution $\mathbf{p}_t = \nabla\tilde{\Phi}(\hat{\mathbf{G}}_{t-1})$.
  Learner receives the reward of chosen arm $g_{t,A_t}$, and estimates reward vector $\hat{\mathbf{g}}_t = \frac{g_{t,A_t}}{p_{t,A_t}}\mathbf{e}_{A_t}$.
  Update : $\hat{\mathbf{G}}_t = \hat{\mathbf{G}}_{t-1} + \hat{\mathbf{g}}_t$.
**end**

**Algorithm 2:** Gradient-Based Prediction Algorithm in Bandit setting

---

If we did not use any smoothing and directly used the baseline potential $\Phi(\mathbf{G}) = \max_{w\in\Delta_{K-1}}\langle w, \mathbf{G}\rangle$, we would be running Follow The Leader (FTL) as our full information algorithm. It is well known that FTL does not have good regret guarantees [13]. Therefore, we need to smooth the baseline potential to induce stability in the algorithm. It turns out that two major algorithm families in online learning, namely Follow The Regularized Leader (FTRL) and Follow The Perturbed Leader (FTPL) correspond to two different types of smoothings.

The smoothing used by FTRL is achieved by adding a strongly convex regularizer in the dual representation of the baseline potential. That is, we set $\tilde{\Phi}(\mathbf{G}) = \mathcal{R}^{\star}(\mathbf{G}) = \max_{w\in\Delta_{K-1}}\langle w, \mathbf{G}\rangle - \eta\mathcal{R}(w)$, where $\mathcal{R}$ is a strongly convex function. The well known exponential weights algorithm [11] uses the Shannon entropy regularizer, $\mathcal{R}_S(w) = \sum_{i=1}^{K} w_i \log(w_i)$. GBPA with the resulting smoothed potential becomes the EXP3 algorithm [7] which achieves a near-optimal regret bound $\mathcal{O}(\sqrt{KT\log K})$ just logarithmically worse compared to the lower bound $\Omega(\sqrt{KT})$. This lower bound was matched by Implicit Normalized Forecaster with polynomial function (Poly-INF algorithm) [4, 5] and later work [2] showed that Poly-INF algorithm is equivalent to FTRL algorithm via the Tsallis entropy regularizer, $\mathcal{R}_{T,\alpha}(w) = \frac{1-\sum_{i=1}^{K} w_i^{\alpha}}{1-\alpha}$ for $\alpha \in (0, 1)$.

An alternate way of smoothing is *stochastic smoothing* which is what is used by FTPL algorithms. It injects stochastic perturbations to the cumulative rewards of each arm and then finds the best arm. Given a perturbation distribution $\mathcal{D}$ and $\mathbf{Z} = (Z_1, \cdots, Z_K)$ consisting of i.i.d. draws from $\mathcal{D}$, the

resulting stochastically smoothed potential is $\tilde{\Phi}(\mathbf{G}; \mathcal{D}) = \mathrm{E}_{Z_1, \cdots, Z_K \sim \mathcal{D}} [\max_{w \in \Delta_{K-1}} \langle w, \mathbf{G} + \eta \mathbf{Z} \rangle]$. Its gradient is $\mathbf{p}_t = \nabla \tilde{\Phi}(\mathbf{G}_t; \mathcal{D}) = \mathrm{E}_{Z_1, \cdots, Z_K \sim \mathcal{D}}[e_{i^\star}] \in \Delta_{K-1}$ where $i^\star = \arg\max_i G_{t,i} + \eta Z_i$.

In Section 4.3, we recall the general regret bound proved by Abernethy et al. [2] for distributions with bounded hazard rate. They showed that a variety of natural perturbation distributions can yield a near-optimal regret bound of $\mathcal{O}(\sqrt{KT \log K})$. However, none of the distributions they tried yielded the minimax optimal rate $\mathcal{O}(\sqrt{KT})$. Since FTRL with Tsallis entropy regularizer can achieve the minimax optimal rate in adversarial bandits, the following is an important unresolved question regarding the power of perturbations.

**Open Problem**  *Is there a perturbation $\mathcal{D}$ such that GBPA with a stochastically smoothed potential using $\mathcal{D}$ achieves the optimal regret bound $\mathcal{O}(\sqrt{KT})$ in adversarial $K$-armed bandits?*

Given what we currently know, there are two very natural approaches to resolving the open question in the affirmative. **Approach 1:** Find a perturbation so that we get the exactly same choice probability function as the one used by FTRL via Tsallis entropy. **Approach 2:** Provide a tighter control on expected block maxima of random variables considered as perturbations by Abernethy et al. [2].

## 4.2  Barrier Against First Approach: Discrete Choice Theory

The first approach is motivated by a folklore observation in online learning theory, namely, that the exponential weights algorithm [11] can be viewed as FTRL via Shannon entropy regularizer or as FTPL via a Gumbel-distributed perturbation. Thus, we might hope to find a perturbation which is an exact equivalent of the Tsallis entropy regularizer. Since FTRL via Tsallis entropy is optimal for adversarial bandits, finding such a perturbation would immediately settle the open problem.

The relation between regularizers and perturbations has been theoretically studied in discrete choice theory [22, 16]. For any perturbation, there is always a regularizer which gives the same choice probability function. The converse, however, does not hold. The Williams-Daly-Zachary Theorem provides a characterization of choice probability functions that can be derived via additive perturbations.

**Theorem 7** (Williams-Daly-Zachary Theorem [22]). *Let $\mathbf{C} : \mathbb{R}^K \to \mathcal{S}_K$ be the choice probability function and derivative matrix $\mathcal{D}\mathbf{C}(\mathbf{G}) = \left( \frac{\partial \mathbf{C}^\intercal}{\partial G_1}, \frac{\partial \mathbf{C}^\intercal}{\partial G_2}, \cdots, \frac{\partial \mathbf{C}^\intercal}{\partial G_K} \right)^\intercal$. The following 4 conditions are necessary and sufficient for the existence of perturbations $Z_i$ such that this choice probability function $\mathbf{C}$ can be written in $C_i(\mathbf{G}) = \mathrm{P}(\arg\max_{j \in [K]} G_j + \eta Z_j = i)$ for $i \in [K]$.*
*(1) $\mathcal{D}\mathbf{C}(\mathbf{G})$ is symmetric, (2) $\mathcal{D}\mathbf{C}(\mathbf{G})$ is positive definite, (3) $\mathcal{D}\mathbf{C}(\mathbf{G}) \cdot \mathbf{1} = 0$, and (4) All mixed partial derivatives of $\mathbf{C}$ are positive, $(-1)^j \frac{\partial^j C_{i_0}}{\partial G_{i_1} \cdots \partial G_{i_j}} > 0$ for each $j = 1, ..., K-1$.*

We now show that if the number of arms is greater than three, there does not exist any perturbation exactly equivalent to Tsallis entropy regularization. Therefore, the first approach to solving the open problem is doomed to failure.

**Theorem 8** (Proof in Appendix A.5). *When $K \geq 4$, there is no stochastic perturbation that yields the same choice probability function as the Tsallis entropy regularizer.*

## 4.3  Barrier Against Second Approach: Extreme Value Theory

The second approach is built on the work of Abernethy et al. [2] who provided the-state-of-the-art perturbation based algorithm for adversarial multi-armed bandits. The framework proposed in this work covered all distributions with bounded hazard rate and showed that the regret of GBPA via perturbation $Z \sim \mathcal{D}$ with a bounded hazard is upper bounded by trade-off between the bound of hazard rate and expected block maxima as stated below.

**Theorem 9** (Theorem 4.2 [2]). *Assume the support of $\mathcal{D}$ is unbounded in positive direction and hazard rate $h_{\mathcal{D}}(x) = \frac{f(x)}{1 - F(x)}$ is bounded, then the expected regret of GBPA($\tilde{\Phi}$) in adversarial bandit is bounded by $\eta \cdot \mathrm{E}[M_K] + \frac{K \sup h_{\mathcal{D}}}{\eta} T$, where $\sup h_{\mathcal{D}} = \sup_{x : f(x) > 0} h_{\mathcal{D}}(x)$. The optimal choice of $\eta$ leads to the regret bound $2\sqrt{KT \cdot \sup h_{\mathcal{D}} \cdot \mathrm{E}[M_K]}$ where $M_K = \max_{i \in [K]} Z_i$.*

Abernethy et al. [2] considered several perturbations such as Gumbel, Gamma, Weibull, Fréchet and Pareto. The best tuning of distribution parameters (to minimize *upper bounds* on the product

$\sup h_{\mathcal{D}} \cdot \mathrm{E}[M_K]$) always leads to the bound $\mathcal{O}(\sqrt{KT \log K})$, which is tantalizingly close to the lower bound but does not match it. It is possible that some of their upper bounds on expected block maxima $\mathrm{E}[M_K]$ are loose and that we can get closer, or perhaps even match, the lower bound by simply doing a better job of bounding expected block maxima (we will not worry about supremum of the hazard since their bounds can easily be shown to be tight, up to constants, using elementary calculations in Appendix B.2). We show that this approach will also not work by *characterizing* the asymptotic (as $K \to \infty$) behavior of block maxima of perturbations using extreme value theory. The statistical behavior of block maxima, $M_K = \max_{i \in [K]} Z_i$, where $Z_i$'s is a sequence of i.i.d. random variables with distribution function $F$ can be described by one of three extreme value distributions: Gumbel, Fréchet and Weibull [8, 23]. Then, the normalizing sequences $\{a_K > 0\}$ and $\{b_K\}$ are explicitly characterized [21]. Under the mild condition, $\mathrm{E}\big((M_K - b_K)/a_K\big) \to \mathrm{E}_{Z \sim G}[Z] = C$ as $K \to \infty$ where $G$ is extreme value distribution and $C$ is constant, and the expected block maxima behave asymptotically as $\mathrm{E}[M_K] = \Theta(C \cdot a_K + b_K)$. See Theorem 11-13 in Appendix B for more details.

Table 1: Asymptotic expected block maxima based on Extreme Value Theory. Gumbel-type and Fréchet-type are denoted by $\Lambda$ and $\Phi_\alpha$ respectively. The Gamma function and the Euler-Mascheroni constant are denoted by $\Gamma(\cdot)$ and $\gamma$ respectively.

| Distribution | Type | $\sup h$ | $\mathrm{E}[M_K]$ |
|---|---|---|---|
| Gumbel($\mu = 0, \beta = 1$) | $\Lambda$ | $1$ | $\log K + \gamma + o(1)$ |
| Gamma($\alpha, 1$) | $\Lambda$ | $1$ | $\log K + \gamma + o(\log K)$ |
| Weibull($\alpha \leq 1$) | $\Lambda$ | $\alpha$ | $(\log K)^{1/\alpha} + o((\log K)^{1/\alpha})$ |
| Fréchet ($\alpha > 1$) | $\Phi_\alpha$ | $\in (\frac{\alpha}{e-1}, 2\alpha)$ | $\Gamma(1 - 1/\alpha) \cdot K^{1/\alpha}$ |
| Pareto($x_m = 1, \alpha$) | $\Phi_\alpha$ | $\alpha$ | $\Gamma(1 - 1/\alpha) \cdot (K^{1/\alpha} - 1)$ |

The asymptotically tight growth rates (with explicit constants for the leading term!) of expected block maximum of some distributions are given in Table 1. They match the upper bounds of the expected block maximum in Table 1 of Abernethy et al. [2]. That is, their upper bounds are asymptotically tight. Gumbel, Gamma and Weibull distribution are Gumbel-type ($\Lambda$) and their expected block maximum behave as $\mathcal{O}(\log K)$ asymptotically. It implies that Gumbel type perturbation can never achieve optimal regret bound despite bounded hazard rate. Fréchet and Pareto distributions are Fréchet-type ($\Phi_\alpha$) and their expected block maximum grows as $K^{1/\alpha}$. Heuristically, if $\alpha$ is set optimally to $\log K$, the expected block maxima is independent of $K$ while the supremum of hazard is upper bounded by $\mathcal{O}(\log K)$.

**Conjecture**  *If there exists a perturbation that achieves minimax optimal regret in adversarial multi-armed bandits, it must be of Fréchet-type.*

Fréchet-type perturbations can still possibly yield the optimal regret bound in perturbation based algorithm if the expected block maximum is asymptotically bounded by a constant and the divergence term in regret analysis of GBPA algorithm can be shown to enjoy a tighter bound than what follows from the assumption of a bounded hazard rate.

**The perturbation equivalent to Tsallis entropy (in two armed setting) is of Fréchet-type** Further evidence to support the conjecture can be found in the connection between FTRL and FTPL algorithms that regularizer $\mathcal{R}$ and perturbation $Z \sim F_{\mathcal{D}}$ are bijective in two-armed bandit in terms of a mapping between $F_{\mathcal{D}^\star}$ and $\mathcal{R}$, $\mathcal{R}(w) - \mathcal{R}(0) = -\int_0^w F_{\mathcal{D}^\star}^{-1}(1-z)dz$, where $Z_1, Z_2$ are i.i.d random variables with distribution function, $F_{\mathcal{D}}$, and then $Z_1 - Z_2 \sim F_{\mathcal{D}^\star}$. The difference of two i.i.d. Fréchet-type distributed random variables is conjectured to be Fréchet-type. Thus, Tsallis entropy in two-armed setting leads to Fréchet-type perturbation, which supports our conjecture about optimal perturbations in adversarial multi-armed bandits. See Appendix C for more details.

## 5   Numerical Experiments

We present some experimental results with perturbation based algorithms (Alg.1-(ii),(iii)) and compare them to the UCB1 algorithm in the simulated stochastic $K$-armed bandit. In all experiments, the number of arms ($K$) is 10, the number of different episodes is 1000, and true mean rewards ($\mu_i$) are generated from $\mathcal{U}[0,1]$ [19]. We consider the following four examples of 1-sub-Gaussian

reward distributions that will be shifted by true mean $\mu_i$; (a) Uniform, $\mathcal{U}[-1, 1]$, (b) Rademacher, $\mathcal{R}\{-1, 1\}$, (c) Gaussian, $\mathcal{N}(0, 1)$, and (d) Gaussian mixture, $W \cdot \mathcal{N}(-1, 1) + (1 - W) \cdot \mathcal{N}(1, 1)$ where $W \sim \text{Bernoulli}(1/2)$. Under each reward setting, we run five different algorithms; UCB1, RCB with Uniform and Rademacher, and FTPL via Gaussian $\mathcal{N}(0, \sigma^2)$ and Double-exponential ($\sigma$) after we use grid search to tune confidence levels for confidence based algorithms and the parameter $\sigma$ for FTPL algorithms. All tuned confidence level and parameter are specified in Figure 1. We compare the performance of perturbation based algorithms to UCB1 in terms of average regret $R(t)/t$, which is expected to more rapidly converge to zero if the better algorithm is used. [1]

The average regret plots in Figure 1 have the similar patterns that FTPL algorithms via Gaussian and Double-exponential consistently perform the best after parameters tuned, while UCB1 algorithm works as well as them in all rewards except for Rademacher reward. The RCB algorithms with Uniform and Rademacher perturbations are slightly worse than UCB1 in early stages, but perform comparably well to UCB1 after enough iterations. In the Rademacher reward case, which is discrete, RCB with Uniform perturbation slightly outperforms UCB1.

Note that the main contribution of this work is to establish theoretical foundations for a large family of perturbation based algorithms (including those used in this section). Our numerical results are not intended to show the superiority of perturbation methods but to demonstrate that they are competitive with Thompson Sampling and UCB. Note that in more complex bandit problems, sampling from the posterior and optimistic optimization can prove to be computationally challenging. Accordingly, our work paves the way for designing efficient perturbation methods in complex settings, such as stochastic linear bandits and stochastic combinatorial bandits, that have both computational advantages and low regret guarantees. Furthermore, perturbation approaches based on the Double-exponential distribution are of special interest from a privacy viewpoint since that distribution figures prominently in the theory of differential privacy [10, 27, 26].

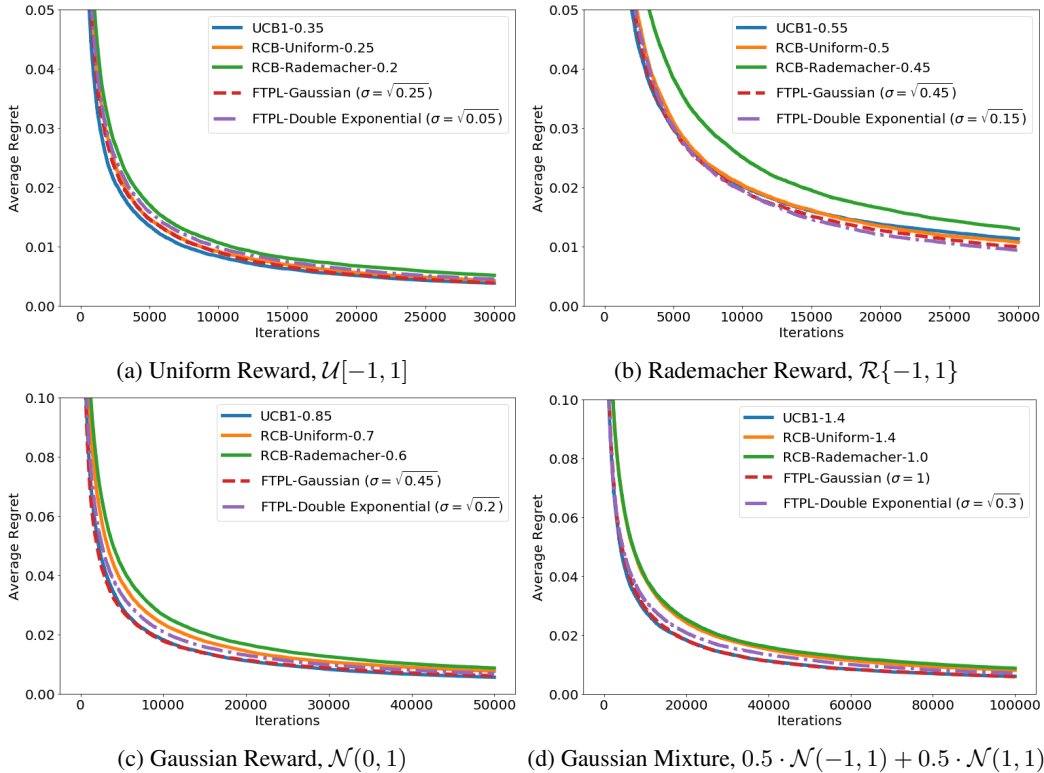

Figure 1: Average Regret for Bandit algorithms in four reward settings (Best viewed in color)

# 6 Conclusion

We provided the first general analysis of perturbations for the stochastic multi-armed bandit problem. We believe that our work paves the way for similar extension for more complex settings, e.g., stochastic linear bandits, stochastic partial monitoring, and Markov decision processes. We also showed that the open problem regarding minimax optimal perturbations for adversarial bandits cannot be solved in two ways that might seem very natural. While our results are negative, they do point the way to a possible affirmative solution of the problem. They led us to a conjecture that the optimal perturbation, if it exists, will be of Fréchet-type.

**Acknowledgments**

We acknowledge the support of NSF CAREER grant IIS-1452099 and the UM-LSA Associate Professor Support Fund. AT was also supported by a Sloan Research Fellowship.

## Footnotes

[1]https://github.com/Kimbaekjin/Perturbation-Methods-StochasticMAB

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
