[Supplementary Material · appendix_camera_ready.pdf]

# A Proofs

## A.1 Proof of Theorem 3

*Proof.* For each arm $i \neq 1$, we will choose two thresholds $x_i = \mu_i + \frac{\Delta_i}{3}, y_i = \mu_1 - \frac{\Delta_i}{3}$ such that $\mu_i < x_i < y_i < \mu_1$ and define two types of events, $E_i^\mu(t) = \{\hat{\mu}_i(t) \leq x_i\}$, and $E_i^\theta(t) = \{\theta_i(t) \leq y_i\}$. Intuitively, $E_i^\mu(t)$ and $E_i^\theta(t)$ are the events that the estimate $\hat{\mu}_i$ and the sample value $\theta_i(t)$ are not too far above the mean $\mu_i$, respectively. $\mathrm{E}[T_i(T)] = \sum_{t=1}^T \mathrm{P}(A_t = i)$ is decomposed into the following three parts according to events $E_i^\mu(t)$ and $E_i^\theta(t)$,

$$\mathrm{E}[T_i(T)] = \underbrace{\sum_{t=1}^T \mathrm{P}(A_t = i, (E_i^\mu(t))^c)}_{(a)} + \underbrace{\sum_{t=1}^T \mathrm{P}(A_t = i, E_i^\mu(t), (E_i^\theta(t))^c)}_{(b)}$$

$$+ \underbrace{\sum_{t=1}^T \mathrm{P}(A_t = i, E_i^\mu(t), E_i^\theta(t))}_{(c)}$$

Let $\tau_k$ denote the time at which $k$-th trial of arm $i$ happens. Set $\tau_0 = 0$.

$$(a) \leq 1 + \sum_{k=1}^{T-1} \mathrm{P}((E_i^\mu(\tau_k + 1))^c) \leq 1 + \sum_{k=1}^{T-1} \exp\left(-\frac{k(x_i - \mu_i)^2}{2}\right) \leq 1 + \frac{18}{\Delta_i^2}. \tag{5}$$

The probability in part $(b)$ is upper bounded by 1 if $T_i(t)$ is less than $L_i(T) = \frac{\sigma^2 [2\log(T\Delta_i^2)]^{2/p}}{(y_i - x_i)^2}$, and by $C_a/(T\Delta_i^2)$ otherwise. The latter can be proved as below,

$$\mathrm{P}(A_t = i, (E_i^\theta(t))^c | E_i^\mu(t)) \leq \mathrm{P}(\theta_i(t) > y_i | \hat{\mu}_i(t) \leq x_i) \leq \mathrm{P}\left(\frac{Z_{it}}{\sqrt{T_i(t)}} > y_i - x_i | \hat{\mu}_i(t) \leq x_i\right)$$

$$\leq C_a \cdot \exp\left(-\frac{T_i(t)^{p/2}(y_i - x_i)^p}{2\sigma^p}\right) \leq \frac{C_a}{T\Delta_i^2} \quad \text{if } T_i(t) \geq L_i(T).$$

The third inequality holds by sub-Weibull $(p)$ assumption on perturbation $Z_{it}$. Let $\tau$ be the largest step until $T_i(t) \leq L_i(T)$, then part $(b)$ is bounded as, $(b) \leq L_i(T) + \sum_{t=\tau+1}^T C_a/(T\Delta_i^2) \leq L_i(T) + C_a/\Delta_i^2$.

Regarding part $(c)$, define $p_{i,t}$ as the probability $p_{i,t} = \mathrm{P}(\theta_1(t) > y_i | \mathcal{H}_{t-1})$ where $\mathcal{H}_{t-1}$ is defined as the history of plays until time $t - 1$. Let $\delta_j$ denote the time at which $j$-th trial of arm 1 happens.

**Lemma 10** (Lemma 1 [3]). *For $i \neq 1$,*

$$(c) = \sum_{t=1}^T \mathrm{P}(A_t = i, E_i^\mu(t), E_i^\theta(t))$$

$$\leq \sum_{t=1}^T \mathrm{E}\left[\frac{1 - p_{i,t}}{p_{i,t}} I(A_t = 1, E_i^\mu(t), E_i^\theta(t))\right] \leq \sum_{j=0}^{T-1} \mathrm{E}\left[\frac{1 - p_{i,\delta_j+1}}{p_{i,\delta_j+1}}\right].$$

*Proof.* See Appendix A.2. $\qquad\square$

The average rewards from the first arm, $\hat{\mu}_1(\delta_j + 1)$, has a density function denoted by $\phi_{\hat{\mu}_{1,j}}$.

$$\mathrm{E}\left[\frac{1 - p_{i,\delta_j+1}}{p_{i,\delta_j+1}}\right] = \mathrm{E}\left[\frac{1}{\mathrm{P}(\theta_1(\delta_j + 1) \geq y_i | \mathcal{H}_{\delta_j+1})} - 1\right]$$

$$= \int_{\mathbb{R}}\left[\frac{1}{\mathrm{P}\left(x + \frac{Z}{\sqrt{j}} > \mu_1 - \frac{\Delta_i}{3}\right)} - 1\right]\phi_{\hat{\mu}_{1,j}}(x)dx$$

The above integration is divided into three intervals, $(-\infty, \mu_1 - \frac{\Delta_i}{3}]$, $(\mu_1 - \frac{\Delta_i}{3}, \mu_1 - \frac{\Delta_i}{6}]$, and $(\mu_1 - \frac{\Delta_i}{3}, \infty)$. We denote them as $(i), (ii)$ and $(iii)$, respectively.

$$\int_{-\infty}^{\mu_1 - \frac{\Delta_i}{3}} \Big[\frac{1}{\mathrm{P}\big(Z > -\sqrt{j}(x - \mu_1 + \frac{\Delta_i}{3})\big)} - C_b\Big]\phi_{\hat{\mu}_{1,j}}(x)dx$$

$$= \int_0^\infty \Big[\frac{1}{\mathrm{P}(Z > u)} - C_b\Big]\frac{1}{\sqrt{j}}\phi_{\hat{\mu}_{1,j}}\big(-\frac{u}{\sqrt{j}} + \mu_1 - \frac{\Delta_i}{3}\big)du \quad \because u = -\sqrt{j}\big(x - \mu_1 + \frac{\Delta_i}{3}\big)$$

$$\leq \int_0^\infty \Big[C_b \cdot \exp\big(\frac{u^q}{2\sigma^q}\big) - C_b\Big]\frac{1}{\sqrt{j}}\phi_{\hat{\mu}_{1,j}}\big(-\frac{u}{\sqrt{j}} + \mu_1 - \frac{\Delta_i}{3}\big)du \quad \because \text{anti-concentration inequality}$$

$$= \int_0^\infty \Big[\int_0^u G'(v)dv\Big]\frac{1}{\sqrt{j}}\phi_{\hat{\mu}_{1,j}}\big(-\frac{u}{\sqrt{j}} + \mu_1 - \frac{\Delta_i}{3}\big)du \quad \because G(u) = C_b \cdot \exp\big(\frac{u^q}{2\sigma^q}\big)$$

$$\leq \int_0^\infty \exp\big(-\frac{(v + \frac{\sqrt{j}\Delta_i}{3})^2}{2}\big) \cdot G'(v)dv \quad \because \text{Fubini's Theorem \& sub-Gaussian reward}$$

$$= \int_0^\infty \exp\big(-\frac{(v + \frac{\sqrt{j}\Delta_i}{3})^2}{2}\big) \cdot C_b \frac{qv^{q-1}}{2\sigma^q}\exp\big(\frac{v^q}{2\sigma^q}\big)dv$$

$$\leq C_b M_{q,\sigma}\exp\big(-\frac{j\Delta_i^2}{18}\big) \quad \because \exists\, 0 < M_{q,\sigma} < \infty \text{ if } q < 2 \text{ or } (q = 2, \sigma \geq 1)$$

$$(i) = \int_{-\infty}^{\mu_1 - \frac{\Delta_i}{3}} \Big[\big[\frac{1}{\mathrm{P}\big(Z > -\sqrt{j}(x - \mu_1 + \frac{\Delta_i}{3})\big)} - C_b\big]\phi_{\hat{\mu}_{1,j}}(x) + (C_b - 1)\phi_{\hat{\mu}_{1,j}}(x)\Big]dx$$

$$\leq C_b M_{q,\sigma}\exp\big(-\frac{j\Delta_i^2}{18}\big) + (C_b - 1)\exp\big(-\frac{j\Delta_i^2}{18}\big)$$

$$(ii) = \int_{\mu_1 - \frac{\Delta_i}{3}}^{\mu_1 - \frac{\Delta_i}{6}} 2\mathrm{P}\big(Z < -\sqrt{j}(x - \mu_1 + \frac{\Delta_i}{3})\big)\phi_{\hat{\mu}_{1,j}}(x)dx$$

$$\leq 2\mathrm{P}(Z < 0) \cdot \mathrm{P}\big(\mu_1 - \frac{\Delta_i}{3} \leq \hat{\mu}_{1,j} \leq \mu_1 - \frac{\Delta_i}{6}\big) \leq 2\mathrm{P}\big(\hat{\mu}_{1,j} \leq \mu_1 - \frac{\Delta_i}{6}\big) \leq 2\exp\big(-\frac{j\Delta_i^2}{72}\big)$$

$$(iii) = \int_{\mu_1 - \frac{\Delta_i}{6}}^\infty 2\mathrm{P}\big(Z < -\sqrt{j}(x - \mu_1 + \frac{\Delta_i}{3})\big)\phi_{\hat{\mu}_{1,j}}(x)dx$$

$$\leq 2\mathrm{P}\big(Z < -\frac{\sqrt{j}\Delta_i}{6}\big)\int_{\mu_1 - \frac{\Delta_i}{6}}^\infty \phi_{\hat{\mu}_{1,j}}(x)dx \leq 2\mathrm{P}\big(Z < -\frac{\sqrt{j}\Delta_i}{6}\big) \leq 2C_a\exp\big(-\frac{j^{p/2}\Delta_i^p}{2 \cdot (6\sigma)^p}\big)$$

$$(c) = \sum_{j=0}^{T-1}(i) + (ii) + (iii) < \frac{18C_b(M_{q,\sigma} + 1) + 126}{\Delta_i^2} + \frac{4C_a(6\sigma)^p}{\Delta_i^p} \tag{6}$$

Combining parts $(a)$, $(b)$, and $(c)$,

$$\mathrm{E}[T_i(T)] \leq 1 + \frac{144 + C_a + 18C_b(M_{q,\sigma} + 1)}{\Delta_i^2} + \frac{4C_a(6\sigma)^p}{\Delta_i^p} + \frac{\sigma^2[2\log(T\Delta_i^2)]^{2/p}}{(y_i - x_i)^2}$$

We obtain the following instance-dependent regret that there exists $C'' = C(\sigma, p, q)$ independent of $K, T$, and $\Delta_i$ such that

$$\mathrm{R}(T) \leq C''\sum_{\Delta_i > 0}\Big(\Delta_i + \frac{1}{\Delta_i} + \frac{1}{\Delta_i^{p-1}} + \frac{\log(T\Delta_i^2)^{2/p}}{\Delta_i}\Big). \tag{7}$$

The optimal choice of $\Delta = \sqrt{K/T}(\log K)^{1/p}$ gives the instance independent regret bound $\mathrm{R}(T) \leq \mathcal{O}(\sqrt{KT}(\log K)^{1/p})$. $\qquad\square$

## A.2 Proof of Lemma 10

*Proof.* First of all, we will show the following inequality holds for all realizations $H_{t-1}$ of $\mathcal{H}_{t-1}$,

$$\mathrm{P}(A_t = i, E_i^\theta(t), E_i^\mu(t)|H_{t-1}) \leq \frac{1 - p_{i,t}}{p_{i,t}} \cdot \mathrm{P}(A_t = 1, E_i^\theta(t), E_i^\mu(t)|H_{t-1}). \tag{8}$$

To prove the above inequality, it suffices to show the following inequality in (9). This is because whether $E_i^\mu(t)$ is true or not depends on realizations $H_{t-1}$ of history $\mathcal{H}_{t-1}$ and we would consider realizations $H_{t-1}$ where $E_i^\mu(t)$ is true. If it is not true in some $H_{t-1}$, then inequality in (8) trivially holds.

$$\mathrm{P}(A_t = i|E_i^\theta(t), H_{t-1}) \leq \frac{1 - p_{i,t}}{p_{i,t}} \cdot \mathrm{P}(A_t = 1|E_i^\theta(t), H_{t-1}) \tag{9}$$

Considering realizations $H_{t-1}$ satisfying $E_i^\theta(t) = \{\theta_i(t) \leq y_i\}$, all $\theta_j(t)$ should be smaller than $y_i$ including optimal arm 1 to choose a sub-optimal arm $i$.

$$\begin{aligned}
\mathrm{P}(A_t = i|E_i^\theta(t), H_{t-1}) &\leq \mathrm{P}(\theta_j(t) \leq y_i, \forall j \in [K]|E_i^\theta(t), H_{t-1}) \\
&= \mathrm{P}(\theta_1(t) \leq y_i|H_{t-1}) \cdot \mathrm{P}(\theta_j(t) \leq y_i, \forall j \in [K] \setminus \{1, i\}|E_i^\theta(t), H_{t-1}) \\
&= (1 - p_{i,t}) \cdot \mathrm{P}(\theta_j(t) \leq y_i, \forall j \in [K] \setminus \{1, i\}|E_i^\theta(t), H_{t-1}) \tag{10}
\end{aligned}$$

The first equality above holds since $\theta_1$ is independent of other $\theta_j, \forall j \neq 1$ and events $E_i^\theta(t)$ given $\mathcal{H}_{t-1}$. In the same way it is obtained as below,

$$\begin{aligned}
\mathrm{P}(A_t = 1|E_i^\theta(t), H_{t-1}) &\geq \mathrm{P}(\theta_1(t) > y_i \geq \theta_j(t), \forall j \in [K] \setminus \{1\}|E_i^\theta(t), H_{t-1}) \\
&= \mathrm{P}(\theta_1(t) \geq y_i|H_{t-1}) \cdot \mathrm{P}(\theta_j(t) \leq y_i, \forall j \in [K] \setminus \{1, i\}|E_i^\theta(t), H_{t-1}) \\
&= p_{i,t} \cdot \mathrm{P}(\theta_j(t) \leq y_i, \forall j \in [K] \setminus \{1, i\}|E_i^\theta(t), H_{t-1}) \tag{11}
\end{aligned}$$

Combining two inequalities (10) and (11), inequality (9) is obtained. The rest of proof is as followed.

$$\begin{aligned}
\sum_{t=1}^T \mathrm{P}(A_t = i, E_i^\mu(t), E_i^\theta(t)) &\leq \sum_{t=1}^T \mathrm{E}[\mathrm{P}(A_t = i, E_i^\mu(t), E_i^\theta(t)|\mathcal{H}_{t-1})] \\
&\leq \sum_{t=1}^T \mathrm{E}\left[\frac{1 - p_{i,t}}{p_{i,t}} \cdot \mathrm{P}(A_t = 1, E_i^\mu(t), E_i^\theta(t)|\mathcal{H}_{t-1})\right] \\
&\leq \sum_{t=1}^T \mathrm{E}\left[\mathrm{E}\left[\frac{1 - p_{i,t}}{p_{i,t}} \cdot \mathrm{I}(A_t = 1, E_i^\mu(t), E_i^\theta(t))|\mathcal{H}_{t-1}\right]\right] \\
&\leq \sum_{t=1}^T \mathrm{E}\left[\frac{1 - p_{i,t}}{p_{i,t}} \cdot \mathrm{I}(A_t = 1, E_i^\mu(t), E_i^\theta(t))\right] \\
&\leq \sum_{j=0}^{T-1} \mathrm{E}\left[\frac{1 - p_{i,\delta_j+1}}{p_{i,\delta_j+1}} \sum_{t=\delta_j+1}^{\delta_{j+1}} \mathrm{I}(A_t = 1, E_i^\mu(t), E_i^\theta(t))\right] \\
&\leq \sum_{j=0}^{T-1} \mathrm{E}\left[\frac{1 - p_{i,\delta_j+1}}{p_{i,\delta_j+1}}\right]
\end{aligned}$$

$\square$

## A.3 Proof of Theorem 5

*Proof.* For each arm $i \neq 1$, we will choose two thresholds $x_i = \mu_i + \frac{\Delta_i}{3}$, $y_i = \mu_1 - \frac{\Delta_i}{3}$ such that $\mu_i < x_i < y_i < \mu_1$ and define three types of events, $E_i^\mu(t) = \{\hat{\mu}_i(t) \leq x_i\}$, $E_i^\theta(t) = \{\theta_i(t) \leq y_i\}$, and $E_{1,i}^\mu(t) = \{\mu_1 - \frac{\Delta_i}{6} - \sqrt{\frac{2\log T}{T_1(t)}} \leq \hat{\mu}_1(t)\}$. The last event is to control the behavior of $\hat{\mu}_1(t)$ not too far below the mean $\mu_1$. $\mathrm{E}[T_i(T)] = \sum_{t=1}^T \mathrm{P}(A_t = i)$ is decomposed into the following four

parts according to events $E_i^\mu(t)$, $E_i^\theta(t)$, and $E_{1,i}^\mu(t)$,

$$\mathrm{E}[T_i(T)] = \underbrace{\sum_{t=1}^{T} \mathrm{P}(A_t = i, (E_i^\mu(t))^c)}_{(a)} + \underbrace{\sum_{t=1}^{T} \mathrm{P}(A_t = i, E_i^\mu(t), (E_i^\theta(t))^c)}_{(b)}$$

$$+ \underbrace{\sum_{t=1}^{T} \mathrm{P}(A_t = i, E_i^\mu(t), E_i^\theta(t), (E_{1,i}^\mu(t))^c)}_{(c)} + \underbrace{\sum_{t=1}^{T} \mathrm{P}(A_t = i, E_i^\mu(t), E_i^\theta(t), E_{1,i}^\mu(t))}_{(d)}.$$

Let $\tau_k$ denote the time at which $k$-th trial of arm $i$ happens. Set $\tau_0 = 0$.

$$(a) \le \mathrm{E}[\sum_{k=0}^{T-1} \sum_{t=\tau_k+1}^{\tau_{k+1}} \mathrm{I}(A_t = i)\mathrm{I}((E_i^\mu(t))^c)] \le \mathrm{E}[\sum_{k=0}^{T-1} \mathrm{I}((E_i^\mu(\tau_k+1))^c) \sum_{t=\tau_k+1}^{\tau_{k+1}} \mathrm{I}(A_t = i)]$$

$$\le 1 + \sum_{k=1}^{T-1} \mathrm{P}((E_i^\mu(\tau_k+1))^c) \le 1 + \sum_{k=1}^{T-1} \exp(-\frac{k(x_i - \mu_i)^2}{2}) \le 1 + \frac{18}{\Delta_i^2}.$$

The second last inequality above holds by Hoeffding bound of sample mean of $k$ sub-Gaussian rewards, $\hat{\mu}_i(t)$ in Lemma 1. The probability in part $(b)$ is upper bounded by 1 if $T_i(t)$ is less than $L_i(T) = \frac{9(2+\epsilon)\log T}{\Delta_i^2}$ and is equal to 0, otherwise. The latter can be proved as below,

$$\mathrm{P}(A_t = i, (E_i^\theta(t))^c | E_i^\mu(t)) \le \mathrm{P}(\theta_i(t) > y_i | \hat{\mu}_i(t) \le x_i)$$

$$\le \mathrm{P}\left(Z_{it} > \sqrt{\frac{T_i(t)(y_i - x_i)^2}{(2+\epsilon)\log T}} | \hat{\mu}_i(t) \le x_i\right) = 0 \quad \text{if} \quad T_i(t) \ge L_i(T).$$

The last equality holds by bounded support of perturbation $Z_{it}$. Let $\tau$ be the largest step until $T_i(t) \le L_i(T)$, then part $(b)$ is bounded by $L_i(T)$. Regarding part $(c)$,

$$(c) = \sum_{t=1}^{T} \mathrm{P}(A_t = i, E_i^\mu(t), E_i^\theta(t), (E_{1,i}^\mu(t))^c)$$

$$\le \sum_{t=1}^{T} \mathrm{P}((E_{1,i}^\mu(t))^c) = \sum_{t=1}^{T} \sum_{s=1}^{T} \mathrm{P}\left(\mu_1 - \frac{\Delta_i}{6} - \sqrt{\frac{2\log T}{s}} \ge \hat{\mu}_{1,s}\right)$$

$$= \sum_{t=1}^{T} \sum_{s=1}^{T} \mathrm{P}\left(\mu_1 - \frac{\Delta_i}{6} \ge \hat{\mu}_{1,s} + \sqrt{\frac{2\log T}{s}}\right) = \sum_{t=1}^{T} \frac{1}{T} \sum_{s=1}^{T} \exp\left(-\frac{s\Delta_i^2}{72}\right) \le \frac{72}{\Delta_i^2}$$

Define $p_{i,t}$ as the probability $p_{i,t} = \mathrm{P}(\theta_1(t) > y_i | \mathcal{H}_{t-1})$ where $\mathcal{H}_{t-1}$ is defined as the history of plays until time $t-1$. Let $\delta_j$ denote the time at which $j$-th trial of arm 1 happens. In the history where the event $E_{1,i}^\mu(t)$ holds, then $\mathrm{P}(\theta_1(t) > y_i | \mathcal{H}_{t-1})$ is strictly greater than zero because of wide

enough support of scaled perturbation by adding an extra logarithmic term in $T$. For $i \neq 1$,

$$(d) = \sum_{t=1}^{T} \mathrm{P}(A_t = i, E_i^{\mu}(t), E_i^{\theta}(t), E_{1,i}^{\mu}(t)) \leq \sum_{j=0}^{T-1} \mathrm{E}\left[\frac{1 - p_{i,\delta_j+1}}{p_{i,\delta_j+1}} \mathrm{I}(E_{1,i}^{\mu}(\delta_j + 1))\right]$$

$$= \sum_{j=0}^{T-1} \mathrm{E}\left[\frac{1 - \mathrm{P}\left(\hat{\mu}_{1,j} + \sqrt{\frac{(2+\epsilon)\log T}{j}} Z \geq \mu_1 - \frac{\Delta_i}{3}\right)}{\mathrm{P}\left(\hat{\mu}_{1,j} + \sqrt{\frac{(2+\epsilon)\log T}{j}} Z \geq \mu_1 - \frac{\Delta_i}{3}\right)} \mathrm{I}\left(\hat{\mu}_{1,j} \geq \mu_1 - \frac{\Delta_i}{6} - \sqrt{\frac{2\log T}{j}}\right)\right]$$

$$= \sum_{j=0}^{T-1} \frac{\mathrm{P}\left(Z \leq \sqrt{\frac{2}{2+\epsilon}} - \frac{\sqrt{j}\Delta_i}{6\sqrt{(2+\epsilon)\log T}}\right)}{\mathrm{P}\left(Z \geq \sqrt{\frac{2}{2+\epsilon}} - \frac{\sqrt{j}\Delta_i}{6\sqrt{(2+\epsilon)\log T}}\right)} \qquad \because \text{maximized when } \hat{\mu}_{1,j} = \mu_1 - \frac{\Delta_i}{6} - \sqrt{\frac{2\log T}{j}}$$

$$= \sum_{j=0}^{M_i(T)} \frac{\mathrm{P}\left(Z \leq \sqrt{\frac{2}{2+\epsilon}} - \frac{\sqrt{j}\Delta_i}{6\sqrt{(2+\epsilon)\log T}}\right)}{\mathrm{P}\left(Z \geq \sqrt{\frac{2}{2+\epsilon}} - \frac{\sqrt{j}\Delta_i}{6\sqrt{(2+\epsilon)\log T}}\right)}$$

$$\leq M_i(T) \cdot \frac{\mathrm{P}\left(Z \leq \sqrt{\frac{2}{2+\epsilon}}\right)}{\mathrm{P}\left(Z \geq \sqrt{\frac{2}{2+\epsilon}}\right)} = M_i(T) \cdot C_{Z,\epsilon} \qquad \because \text{maximized when } j = 0$$

The first inequality holds by Lemma 10, and the last equality works since the term inside expectation becomes zero if $j \geq M_i(T) = \left(36(\sqrt{2} + \sqrt{(2+\epsilon)})^2 \log T\right)/\Delta_i^2$. This is because the lower bound of perturbed average rewards from the arm 1 becomes larger than $y_i$ for $j \geq M_i(T)$. Combining parts $(a)$, $(b)$, $(c)$ and $(d)$,

$$\mathrm{E}[T_i(T)] \leq 1 + \frac{90}{\Delta_i^2} + \frac{9(2+\epsilon)\log T}{\Delta_i^2} + C_{Z,\epsilon} \cdot \frac{36(\sqrt{2} + \sqrt{(2+\epsilon)})^2 \log T}{\Delta_i^2}$$

Thus, the instance-dependent regret bound is obtained as below, there exist a universal constant $C''' > 0$ independent of $T$, $K$ and $\Delta_i$,

$$\mathrm{R}(T) = C''' \sum_{\Delta_i > 0} \left(\Delta_i + \frac{\log(T)}{\Delta_i}\right).$$

The optimal choice of $\Delta = \sqrt{K \log T / T}$, the instance-independent regret bound is derived as it follows,

$$\mathrm{R}(T) \leq \mathcal{O}(\sqrt{KT \log T})$$

$\square$

## A.4 Proof of Theorem 6

*Proof.* The proof is a simple extension of the work of Agrawal and Goyal [3]. Let $\mu_1 = \Delta = \sqrt{K/T}(\log K)^{1/q}, \mu_2 = \mu_3 = \cdots = \mu_K = 0$ and each reward is generated from a point distribution. Then, sample means of rewards are $\hat{\mu}_1(t) = \Delta$ and $\hat{\mu}_i(t) = 0$ if $i \neq 1$. The normalized $\theta_i(t)$ sampled from the FTPL algorithm is distributed as $\sqrt{T_i(t)} \cdot (\theta_i(t) - \hat{\mu}_i(t)) \sim Z$.

Define the event $E_{t-1} = \{\sum_{i \neq 1} T_i(t) \leq c\sqrt{KT}(\log K)^{1/q}/\Delta\}$ for a fixed constant $c$. If $E_{t-1}$ is not true, then the regret until time $t$ is at least $c\sqrt{KT}(\log K)^{1/q}$. For any $t \leq T$, $\mathrm{P}(E_{t-1}) \leq 1/2$. Otherwise, the expected regret until time $t$, $\mathrm{E}[R(t)] \geq \mathrm{E}[R(t)|E_{t-1}^c] \cdot 1/2 = \Omega(\sqrt{KT}(\log K)^{1/q})$. If $E_{t-1}$ is true, the probability of playing a suboptimal is at least a constant, so that regret is $\Omega(T\Delta) = \Omega(\sqrt{KT}(\log K)^{1/q})$.

$$\mathrm{P}(\exists i \neq 1, \theta_i(t) > \mu_1 | \mathcal{H}_{t-1}) = \mathrm{P}(\exists i \neq 1, \theta_i(t)\sqrt{T_i(t)} > \Delta\sqrt{T_i(t)} | \mathcal{H}_{t-1})$$

$$= \mathrm{P}(\exists i \neq 1, Z > \Delta\sqrt{T_i(t)} | \mathcal{H}_{t-1})$$

$$\geq 1 - \prod_{i \neq 1}\left(1 - \exp\left(-(\sqrt{T_i(t)}\Delta/\sigma)^q/2\right)/C_b\right)$$

Given realization $H_{t-1}$ of history $\mathcal{H}_{t-1}$ such that $E_{t-1}$ is true, we have $\sum_{i \neq 1} T_i(t) \leq \frac{c\sqrt{KT}(\log K)^{1/q}}{\Delta}$ and it is minimized when $T_i(t) = \frac{c\sqrt{KT}(\log K)^{1/q}}{(K-1)\Delta}$ for all $i \neq 1$. Then,

$$P(\exists i \neq 1, \theta_i(t) > \mu_1 | H_{t-1}) \geq 1 - \prod_{i \neq 1} \left( 1 - \exp\left( - \frac{(\sqrt{T_i(t)}\Delta)^q}{2\sigma^q} \right)/C_b \right)$$

$$= 1 - \left( 1 - \frac{\sigma(q, K)}{K} \right)^{K-1}$$

where $\sigma(q, K) = \exp\left( \frac{c^{q/2}}{2\nu^q}(\frac{K}{K-1})^{q/2} \right)/C_b$. Accordingly,

$$P(\exists i \neq 1, A_t = i) \geq \frac{1}{2}\left( 1 - \left( 1 - \frac{\sigma(q, K)}{K} \right)^{K-1} \right) \cdot \frac{1}{2} \to p^\star \in (0, 1).$$

Therefore, the regret in time $T$ is at least $Tp^\star\Delta = \Omega(\sqrt{KT}(\log K)^{1/q})$.  $\square$

### A.5  Proof of Theorem 8

*Proof.* Fix $\eta = 1$ without loss of generality in FTRL algorithm via Tsallis entropy. For any $\alpha \in (0, 1)$, Tsallis entropy yields the following choice probability, $C_i(\mathbf{G}) = \left( \frac{1-\alpha}{\alpha} \right)^{\frac{1}{\alpha-1}} (\lambda(\mathbf{G}) - G_i)^{\frac{1}{\alpha-1}}$, where $\sum_{i=1}^K C_i(\mathbf{G}) = 1$, $\lambda(\mathbf{G}) \geq \max_{i \in [K]} G_i$. Then for $1 \leq i \neq j \leq K$, the first derivative is negative as shown below,

$$\frac{\partial C_i(\mathbf{G})}{\partial G_j} = \left( \frac{1-\alpha}{\alpha} \right)^{\frac{1}{\alpha-1}} \frac{1}{\alpha-1} \frac{(\lambda(\mathbf{G}) - G_i)^{\frac{1}{\alpha-1}-1}(\lambda(\mathbf{G}) - G_j)^{\frac{1}{\alpha-1}-1}}{\sum_{l=1}^K (\lambda(\mathbf{G}) - G_l)^{\frac{1}{\alpha-1}-1}} < 0.$$

and it implies that $\mathcal{D}\mathbf{C}(\mathbf{G})$ is symmetric. For, $1 \leq i \neq j \neq k \leq K$, the second partial derivative, $\frac{\partial^2 C_i(\mathbf{G})}{\partial G_j \partial G_k}$ is derived as

$$C_i(\mathbf{G}) \cdot \left( \left( \sum_{l=1}^K (\lambda(\mathbf{G}) - G_l)^{\frac{2-\alpha}{\alpha-1}} \right) \left( \frac{1}{\lambda(\mathbf{G}) - G_i} + \frac{1}{\lambda(\mathbf{G}) - G_j} + \frac{1}{\lambda(\mathbf{G}) - G_k} \right) - \sum_{l=1}^K (\lambda(\mathbf{G}) - G_l)^{\frac{3-2\alpha}{\alpha-1}} \right). \tag{12}$$

If we set $1/(\lambda(\mathbf{G}) - G_i) = X_i$, the term in (12) except for the term $C_i(\mathbf{G})$ is simplified to $\sum_{i=1}^K X_i^{\frac{2-\alpha}{1-\alpha}} \cdot (X_1 + X_2 + X_3) - \sum_{i=1}^K X_i^{\frac{3-2\alpha}{1-\alpha}}$ where $\sum_{i=1}^K X_i^{\frac{1}{1-\alpha}} = \left( \frac{\alpha}{1-\alpha} \right)^{\frac{1}{\alpha-1}}$. If we set $K = 4$, $C(\mathbf{G}) = (\epsilon, \epsilon, \epsilon, 1 - 3\epsilon)$, and $X_i = C_i^{1-\alpha}\frac{1-\alpha}{\alpha}$, then it is equal to

$$\left( \frac{1-\alpha}{\alpha} \right)^{\frac{3-2\alpha}{1-\alpha}} \left( 6\epsilon^{3-2\alpha} + 3\epsilon^{1-\alpha}(1 - 3\epsilon)^{2-\alpha} - (1 - 3\epsilon)^{3-2\alpha} \right). \tag{13}$$

So, there always exists $\epsilon > 0$ small enough to make the value of (13) negative where $\alpha \in (0, 1)$, which means condition (4) in Theorem 7 is violated.  $\square$

## B  Extreme Value Theory

### B.1  Extreme Value Theorem

**Theorem 11** (Proposition 0.3 [23])**.** *Suppose that there exist $\{a_K > 0\}$ and $\{b_K\}$ such that*

$$P((M_K - b_K)/a_K \leq z) = F^K(a_K \cdot z + b_K) \to G(z) \quad \text{as } K \to \infty \tag{14}$$

*where $G$ is a non-degenerate distribution function, then $G$ belongs to one of families; Gumbel, Fréchet and Weibull. Then, $F$ is in the domain of attraction of $G$, written as $F \in D(G)$.*
*1. Gumbel type ($\Gamma$) with $G(x) = \exp(-\exp(-x))$ for $x \in \mathbb{R}$.*
*2. Fréchet type ($\Phi_\alpha$) with $G(x) = 0$ for $x > 0$ and $G(x) = \exp(-x^{-\alpha})$ for $x \geq 0$.*
*3. Weibull type ($\Psi_\alpha$) with $G(x) = \exp(-(-x)^\alpha)$ for $x \leq 0$ and $G(x) = 1$ for $x > 0$.*

Let $\gamma_K = F^{\leftarrow}(1 - 1/K) = \inf\{x : F(x) \geq 1 - 1/K\}$.

**Theorem 12** (Proposition 1.1 [23])**.** *Type 1 - Gumbel ($\Lambda$)*

*1. If $F \in D(\Gamma)$, there exists some strictly positive function $g(t)$ s.t. $\lim_{t \to \omega_F} \frac{1-F(t+x \cdot g(t))}{1-F(t)} = \exp(-x)$ for all $x \in \mathbb{R}$ with exponential tail decay. Its corresponding normalizing sequences are $a_K = g(\gamma_K)$ and $b_K = \gamma_K$, where $g = (1 - F)/F'$.*

*2. If $\lim_{x \to \infty} \frac{F''(x)(1-F(x))}{\{F'(x)\}^2} = -1$, then $F \in D(\Lambda)$.*

*3. If $\int_{-\infty}^{0} |x| F(dx) < \infty$, then $\lim_{K \to \infty} \mathrm{E}\big((M_K - b_K)/a_K\big) = -\Gamma^{(1)}(1)$. Accordingly, $\mathrm{E}M_K$ behaves as $-\Gamma^{(1)}(1) \cdot g(\gamma_K) + \gamma_K$.*

**Theorem 13** (Proposition 1.11 [23])**.** *Type 2 - Fréchet ($\Phi_\alpha$)*

*1. If $F \in D(\Phi_\alpha)$, its upper end point is infinite, $\omega_F = \infty$, and it has tail behavior that decays polynomially $\lim_{t \to \infty} \frac{1-F(tx)}{1-F(t)} = x^{-\alpha}$, for $x > 0, \alpha > 0$. Its corresponding normalizing sequences are $a_K = \gamma_K$ and $b_K = 0$.*

*2. If $\lim_{x \to \infty} \frac{x F'(x)}{1-F(x)} = \alpha$ for some $\alpha > 0$, then $F \in D(\Phi_\alpha)$.*

*3. If $\alpha > 1$ and $\int_{-\infty}^{0} |x| F(dx) < \infty$, then $\lim_{K \to \infty} \mathrm{E}\big(M_K/a_K\big) = \Gamma\big(1 - 1/\alpha\big)$. Accordingly, $\mathrm{E}M_K$ behaves as $\Gamma\big(1 - \frac{1}{\alpha}\big) \cdot \gamma_K$.*

## B.2 Asymptotic Expected Block Maxima and Supremum of Hazard Rate

### B.2.1 Gumbel distribution

Gumbel has the following distribution function, the first derivative and the second derivative, $F(x) = \exp(-e^{-x})$, $F'(x) = e^{-x}F(x)$, and $F''(x) = (e^{-x} - 1)F'(x)$. $\lim_{x \to \infty} \frac{F''(x)(1-F(x))}{\{F'(x)\}^2} = -1$, thus this is Gumbel-type distribution by Theorem 12, $F \in D(\Lambda)$. If $g(x) = e^x(e^{e^{-x}} - 1)$, then normalizing constants are obtained as $b_K = -\log(-\log(1 - 1/K)) \sim \log K$, and $a_K = g(b_K) = (1 - F(b_K))/(\exp(-b_K)F(b_K)) = 1 + 1/K + o(\frac{1}{K})$. Accordingly, $\mathrm{E}M_K = -\Gamma^{(1)}(1) \cdot (1 + \frac{1}{K}) + \log K + o(1/K)$.

Its hazard rate is derived as $h(x) = \frac{F'(x)}{1-F(x)} = \frac{e^{-x}}{\exp(e^{-x})-1}$, and since it increases monotonically and converges to 1 as $x$ goes to infinity, it has an asymptotically tight bound 1.

### B.2.2 Gamma distribution

For $x > 0$, the first derivative and the second derivative of distribution function are given as $F'(x) = (x^{\alpha-1}e^{-x})/\Gamma(\alpha)$ and $F''(x) = -F'(x)(1 + (\alpha - 1)/t) \sim -F'(x)$. It satisfies $\frac{F''(1-F(x))}{(F'(x))^2} \sim -\frac{1-F(x)}{F'(x)} \to -1$ so it is Gumbel-type by Theorem 12, $F \in D(\Lambda)$. It has $g(x) \to 1$ and thus $a_K = 1$. Since $F'(b_K) \sim 1 - F(b_K) = 1/K$, $(\alpha - 1)\log b_K - b_K - \log\Gamma(\alpha) = -\log K$. Thus, we have $b_K = \log K + (\alpha - 1)\log\log K - \log\Gamma(\alpha)$. Accordingly, $\mathrm{E}M_K = -\Gamma^{(1)}(1) + \log K + (\alpha - 1)\log\log K - \log\Gamma(\alpha)$.

Its hazard function is expressed by $h(x) = (x^{\alpha-1}\exp(-x))/[\int_x^\infty t^{\alpha-1}\exp(-t)dt]$. It increases monotonically and converges to 1, and thus has an asymptotically tight bound 1.

### B.2.3 Weibull distribution

The Weibull distribution function and its first derivative are obtained as as $F(x) = 1 - \exp(-(x + 1)^\alpha + 1)$ and $F'(x) = \alpha(x+1)^{\alpha-1}(1 - F(x))$. Its second derivative is $(\frac{\alpha-1}{x+1} - \alpha(x+1)^{\alpha-1}) \cdot F'(x)$. The second condition in Theorem 12 is satisfied, and thus $F \in D(\Lambda)$ and $g(x) = x^{-\alpha+1}/\alpha$. Corresponding normalizing constants are derived as $b_K = (1 + \log K)^{1/\alpha} - 1 \sim (\log K)^{1/\alpha}$ and $a_K = g(b_K) = (\log K)^{1/\alpha-1}/\alpha$. So, $\mathrm{E}M_K = -\Gamma^{(1)}(1) \cdot (\log K)^{1/\alpha-1}/\alpha + (\log K)^{1/\alpha}$.

Its hazard rate function is $h(x) = \alpha(x + 1)^{\alpha-1}$ for $x \geq 0$. If $\alpha > 1$, it increases monotonically and becomes unbounded. If the case for $\alpha \leq 1$ is only considered, then the hazard rate is tightly bounded by $\alpha$.

### B.2.4 Fréchet distribution

The first derivative of Fréchet distribution function is $F'(x) = \exp(-x^{-\alpha})\alpha x^{-\alpha-1}$ for $x > 0$ and the second condition in Theorem 13 is satisfied as $\lim_{x\to\infty} \frac{xF'(x)}{1-F(x)} = \lim_{x\to\infty} \frac{\alpha x^{-\alpha}}{\exp(x^{-\alpha})-1} \to \alpha$. Thus, it is Fréchet-type distribution ($\Phi_\alpha$) so that $b_K = 0$ and $a_K = [-\log(1-1/K)]^{-1/\alpha} = [1/K + o(1/K)]^{-1/\alpha} \sim K^{1/\alpha}$. So, $\mathrm{E}M_K = \Gamma(1-1/\alpha) \cdot K^{1/\alpha}$.

The hazard rate is $h(x) = \alpha x^{-\alpha-1} \frac{1}{\exp(x^{-\alpha})-1}$. It is already known that supremum of hazard is upper bound by $2\alpha$ in Appendix D.2.1 in Abernethy et al. [2]. Regarding the lower bound of a hazard rate, $\sup_{x>0} h(x) \geq h(1) = \alpha/(e-1)$.

### B.2.5 Pareto distribution

The modified Pareto distribution function is $F(x) = 1 - \frac{1}{(1+x)^\alpha}$ for $x \geq 0$. The second condition in Theorem 13 is met as $\lim_{x\to\infty} \frac{xF'(x)}{1-F(x)} = \lim_{x\to\infty} \frac{\alpha x}{1+x} \to \alpha > 1$. Thus, it is Fréchet-type distribution ($\Phi_\alpha$), and has normalizing constants, $b_K = 0$ and $a_K = K^{1/\alpha} - 1$. Accordingly, $\mathrm{E}M_K \approx \Gamma(1-1/\alpha) \cdot (K^{1/\alpha} - 1)$.

Its hazard rate is $h(x) = \frac{\alpha}{1+x}$ for $x \geq 0$ so that it is tightly bounded by $\alpha$.

## C  Two-armed Bandit setting

### C.1  Shannon entropy

There is a mapping between $\mathcal{R}$ and $F_{\mathcal{D}^\star}$,

$$\mathcal{R}(w) - \mathcal{R}(0) = -\int_0^w F_{\mathcal{D}^\star}^{-1}(1-z)dz. \tag{15}$$

Let $\mathcal{R}(w)$ be one-dimensional Shannon entropic regularizer, $\mathcal{R}(w) = -w\log w - (1-w)\log(1-w)$ for $w \in (0,1)$ and its first derivative is $\mathcal{R}'(w) = \log\frac{1-w}{w} = F_{\mathcal{D}^\star}^{-1}(1-w)$. Then $F_{\mathcal{D}^\star}(z) = \frac{\exp(z)}{1+\exp(z)}$, which can be interpreted as the difference of two Gumbel distribution as follows,

$$\mathrm{P}(\arg\max_{w\in\Delta_1}\langle w, (G_1+Z_1, G_2, +Z_2)\rangle = 1) = \mathrm{P}(G_1 + Z_1 > G_2 + Z_2))$$
$$= \mathrm{P}(Y > G_2 - G_1) \quad \text{where } Y = Z_1 - Z_2 \sim \mathcal{D}^\star$$
$$= 1 - F_{\mathcal{D}^\star}(G_2 - G_1) = \frac{\exp(G_1)}{\exp(G_1) + \exp(G_2)}$$

If $Z_1, Z_2 \sim Gumbel(\alpha, \beta)$ and are independent, then $Z_1 - Z_2 \sim Logistic(0, \beta)$. Therefore, the perturbation, $F_{\mathcal{D}^\star}$ is not distribution function for Gumbel, but Logistic distribution which is the difference of two i.i.d Gumbel distributions. Interestingly, the logistic distribution turned out to be also Gumbel types extreme value distribution as Gumbel distribution. It is naturally conjectured that the difference between two i.i.d Gumbel types distribution with exponential tail decay must be Gumbel types as well. The same holds for Fréchet-type distribution with polynomial tail decay.

### C.2  Tsallis entropy

Theorem 8 states that there does not exist a perturbation that gives the choice probability function same as that from FTRL via Tsallis entropy when $K \geq 4$. In two-armed setting, however, there exists a perturbation equivalent to Tsallis entropy and this perturbation naturally yields an optimal perturbation based algorithm.

Let us consider Tsallis entropy regularizer in one dimensional decision set expressed by $R(w) = \frac{1}{1-\alpha}(-1 + w^\alpha + (1-w)^\alpha)$ for $w \in (0,1)$ and its first derivative is $R'(w) = \frac{\alpha}{1-\alpha}(w^{\alpha-1} - (1-w)^{\alpha-1}) = F_{\mathcal{D}^\star}^{-1}(1-w)$. If we set $u = 1-w$, then the implicit form of distribution function and density function are given as $F_{\mathcal{D}^\star}(\frac{\alpha}{1-\alpha}((1-u)^{\alpha-1} - u^{\alpha-1})) = u$ and $f_{\mathcal{D}^\star}(\frac{\alpha}{1-\alpha}((1-u)^{\alpha-1} - u^{\alpha-1})) = \frac{1}{\alpha((1-u)^{\alpha-2}+u^{\alpha-2})}$. As $u$ converges to 1, then $z = \frac{\alpha}{1-\alpha}((1-u)^{\alpha-1} - u^{\alpha-1})$ goes to positive infinity.

This distribution satisfies the second condition in Theorem 13 so that it turns out to be Fréchet-type.

$$\lim_{z \to \infty} \frac{z f_{\mathcal{D}^\star}(z)}{1 - F_{\mathcal{D}^\star}(z)} = \lim_{u \to 1} \frac{\frac{\alpha}{1-\alpha}((1-u)^{\alpha-1} - u^{\alpha-1})}{(1-u) \times \alpha((1-u)^{\alpha-2} + u^{\alpha-2})} = \frac{1}{1-\alpha}.$$

If the conjecture above holds, the optimal perturbation that corresponds to Tsallis entropy regularizer must be also Fréchet-type distribution in two armed bandit setting. This result strongly support our conjecture that the perturbation in an optimal FTPL algorithm must be Fréchet-type.