[Reviews · NeurIPS 2019]

Reviewer 1



The paper studies the Follow-the-perturbed (FTPL) leader algorithms for multi-armed bandit problems (both stochastic and adversarial). In the stochastic setting, a unified analysis of FTPL with sub-Weibull perturbations and FTPL with bouded perturbatinos is given in the case of sub-Gaussian arms. In the adversarial setting, the paper provide a negative result regarding the construction of a FTPL algorithm achieving the optimal $\sqrt{KT}$ regret bound: it asserts that there is no perturbation so that FTPL is equivalent to the optimal INF algorithm. A conjecture, based on extreme-value theory, regarding the form a such an optimal perturbation is proposed. Overall, I believe the paper brings several interesting contributions to the study of FTPL algorithms for MAB.

Reviewer 2



********* [added after the rebuttal] I have read the reviews of the other reviewers, who did not raise any major concerns, and so the authors' rebuttal wasn't expected to bring any major correction or clarification. Overall I stick to my originally given score. ********* Originality: for the adversarial setting, the paper builds on and significantly extends previous work [1,2]. For the stochastic setting, it seems to be novel. Quality: technically strong, using concepts and methods from different theories, providing several statements with proofs and a conjecture. Clarity: very good, easy to read. Significance: I found the unification ideas of high interest. The problem in the adversarial setting is very hard and the authors showed a mix of negative results and a positive result in a special case. Minor comments: L52: please specify the range for t (starting at 0 or 1?) L83: after -> before L167: FTRL is not defined until L185 L285: what is the difference between the number of epizodes and the number of iterations (mentioned in Figure 1)? L287: define the algorithm parameters that need tuning L288: B(1,1/2) is not defined

Reviewer 3



Originality: This paper manages to unify different perturbation based algorithms to one template and as a byproduct it gives a randomized version of UCB. Moreover it excludes the two natural approaches for obtaining the optimal regret bound for adversarial bandit problem and provide a conjecture with strong evidence. Quality: The proofs are reliable. The paper gives supporting argument on the conjecture it proposes. Clarity: The paper is clearly written. Significance: The general analysis on perturbed algorithms for stochastic multi-armed bandit problem is inspiring. We may use it for more complex settings. Also it eliminates two possible ways of pursuing the optimal regret bound for adversarial multi-armed bandit problem and shine a light on what the optimal perturbation should be.

[Author Response · NeurIPS 2019]

We really appreciate the time and expertise you have invested in these reviews. We wish to express our appreciation for your in-depth comments, suggestions, and corrections, which will greatly improve the manuscript. We will reply to individual questions from reviewers respectively. Note that the numbers in the numbered lists below refer to sections of the review form as follows: 1=Contributions, 2=Detailed Comments, 5=Improvements.

**Reviewer #1**

1. We agree with your summary of our contributions and would like to emphasize again that we generalized the regret analysis of existing Gaussian Thompson Sampling [3] in two significant respects; (1) from the special Gaussian perturbation to general sub-Weibull or bounded perturbations, and (2) from the special Gaussian rewards distribution to general sub-Gaussian rewards. We would also like to emphasize that the lower bound in the stochastic case (Theorem 6) means that the regret analysis in important specific cases, like the Gaussian and double exponential perturbations, is tight.
2. We concur and this is an accurate summary for both settings in this work.
5. We agree and are working on the project of solving this open problem in the adversarial bandit setting. Our negative results around barriers to natural approaches to solving the open problem do have a positive aspect: they will save future researchers from spending time in these fruitless directions (as we did until we ran into these provable barriers).

**Reviewer #2**

1. We concur with your description of the main contributions of our paper for stochastic and adversarial bandit settings.
2. Thanks for your acknowledgement that the adversarial problem is really hard. It is this realization that makes even partial progress (in the form of barrier results) worth publishing. Thanks so much for detailed comments. We will fix them resulting in an improved manuscript.
5. Same answer as in point #5 for Reviewer #1 above.

**Reviewer #3**

1. We agree that this is an accurate summary of our contributions in both stochastic and adversarial bandit settings.
2. We agree. As you mention in significance part, our work paves the way for the design and analysis of efficient perturbation algorithms that enjoys both computational advantages and low regret guarantees in more complex settings such as stochastic linear bandits, combinatorial bandits and partial monitoring games.
5. In the paragraph "Failure of Bounded Perturbation" (L134-L141), we provided a counterexample that a perturbation algorithm via Uniform distribution but without log term will achieve a linear regret in two armed bandit problem. As an arm is pulled several times, the width of perturbation gets smaller because of scaling term $(1/\sqrt{T_i(t)})$, and thus the ranges of sampling distributions from two arms stop being overlapping so that the algorithm stops exploring and incurs linear regret. Therefore, we clearly need to add a term which is an *increasing* function of "global" time $T$ so that it can compensate for narrow sampling range and restores good regret properties of the algorithm by increasing the width of perturbation. It is less easy to motivate in non-technical terms why the the term has to be *logarithmic* in time. Perhaps an analogy will help: the logarithmic term in the numerator also appears in optimistic algorithms like UCB. So it is satisfying that the randomized version randomizes within an interval of the same scaling over which UCB optimizes.

[Meta-Review · NeurIPS 2019]

All the reviewers agree that the paper contributes novel results on the role of perturbations in multi armed bandit problems, of both the adversarial and stochastic varieties. Specifically, it provides a unified, Follow-the-Perturbed-Leader viewpoint for studying different perturbation based approaches to bandits, and as a byproduct shows new and well-performing variations of standard algorithms such as UCB. This is likely to be of interest in the theoretical design of online learning algorithms, and its ideas can be potentially extended to more complicated bandit settings with structure.